# NanoDefiner Framework and e-Tool Revisited According to the European Commission’s Nanomaterial Definition 2022/C 229/01

**DOI:** 10.3390/nano13060990

**Published:** 2023-03-09

**Authors:** Raphael Brüngel, Johannes Rückert, Philipp Müller, Frank Babick, Christoph M. Friedrich, Antoine Ghanem, Vasile-Dan Hodoroaba, Agnieszka Mech, Stefan Weigel, Wendel Wohlleben, Hubert Rauscher

**Affiliations:** 1Department of Computer Science, University of Applied Sciences and Arts Dortmund (FH Dortmund), 44227 Dortmund, Germany; 2Institute for Medical Informatics, Biometry and Epidemiology (IMIBE), University Hospital Essen, 45122 Essen, Germany; 3Institute for Artificial Intelligence in Medicine (IKIM), University Hospital Essen, 45131 Essen, Germany; 4Analytical and Material Science, BASF SE, 67056 Ludwigshafen, Germany; 5Institute of Process Engineering and Environmental Technology, Technische Universität Dresden (TU Dresden), 01062 Dresden, Germany; 6R&I Centre Brussels, Solvay S.A., 1120 Brussels, Belgium; 7Division 6.1 Surface Analysis and Interfacial Chemistry, Bundesanstalt für Materialforschung und -prüfung (BAM), 12205 Berlin, Germany; 8European Commission, Joint Research Centre (JRC), 21027 Ispra, Italy; 9Department Safety in the Food Chain, German Federal Institute for Risk Assessment (BfR), 10589 Berlin, Germany

**Keywords:** nanomaterial definition, nanomaterial categorisation, nanomaterial regulation, nanomaterial legislation, decision support, FAIRification

## Abstract

The new recommended definition of a nanomaterial, 2022/C 229/01, adopted by the European Commission in 2022, will have a considerable impact on European Union legislation addressing chemicals, and therefore tools to implement this new definition are urgently needed. The updated NanoDefiner framework and its e-tool implementation presented here are such instruments, which help stakeholders to find out in a straightforward way whether a material is a nanomaterial or not. They are two major outcomes of the NanoDefine project, which is explicitly referred to in the new definition. This work revisits the framework and e-tool, and elaborates necessary adjustments to make these outcomes applicable for the updated recommendation. A broad set of case studies on representative materials confirms the validity of these adjustments. To further foster the sustainability and applicability of the framework and e-tool, measures for the FAIRification of expert knowledge within the e-tool’s knowledge base are elaborated as well. The updated framework and e-tool are now ready to be used in line with the updated recommendation. The presented approach may serve as an example for reviewing existing guidance and tools developed for the previous definition 2011/696/EU, particularly those adopting NanoDefine project outcomes.

## 1. Introduction

The European Commission (EC) has adopted a new recommendation for the overarching definition of the term “nanomaterial”, 2022/C 229/01 [1], which updates the previous recommendation, 2011/696/EU [2], as of 10 June 2022. This update aims to simplify the understanding of the term “nanomaterial” and to improve its implementability in a regulatory context, taking into account comprehensive reviews on experiences [3], evaluations [4], and scientific-technical options [5]. In the following, the definition 2011/696/EU, published in 2011, will be called the “2011-Recommendation”, whereas the new definition 2022/C 229/01, published in 2022, will be called the “2022-Recommendation”.

The 2022-Recommendation refers to the NanoDefine project (https://web.archive.org/web/20221128020814/http://nanodefine.eu/ (accessed 24 February 2023); https://cordis.europa.eu/project/id/604347 (accessed 30 November 2022)) and its outcomes, summarised within the NanoDefine Methods Manual [6,7,8], when elaborating the technical and scientific elements that underpinned the 2011-Recommendation’s review. The NanoDefine project was one of the European Union’s (EU) Framework Programme 7 (FP7) flagship projects, investigating open questions that resulted from the practical implementation of the 2011-Recommendation. In a nutshell, NanoDefine aimed to provide guidance and develop methods for the determination of particle size, which is the key parameter for identifying nanomaterials. A central part of the NanoDefine outcomes is the NanoDefiner framework for the categorisation of potential nanomaterials. The framework incorporates a decision support flow scheme [6,9] for guidance, and its implementation as expert system software, i.e., the NanoDefiner e-tool [6,10]. Other EU projects and institutions have relied on these outcomes as well, e.g., the Horizon 2020 (H2020) ACEnano project (http://www.acenano-project.eu/ (accessed 30 November 2022)), which developed tools (http://www.acenano-project.eu/acenano-toolbox (accessed 30 November 2022)) for nanomaterial risk assessment, and the European Centre for Ecotoxicology and Toxicology of Chemicals (ECETOC) (https://www.ecetoc.org/ (accessed 30 November 2022)) with its NanoApp (https://nanoapp.ecetoc.org/ (access 30 November 2022)) [11,12], which is able to use e-tool decisions and output.

The 2011-Recommendation was integrated into several pieces of EU legislation, and, therefore, guidance addressing the 2011-Recommendation and tools developed on this basis are still valid as long as it is relevant for legislation. However, the 2022-Recommendation replaces the old one and will continuously gain regulatory importance over a transition period, during which the new definition is expected to be incorporated in legislation. Hence, the respective guidance and tools (more than 500 tools inventoried in the NANoREG Toolbox [13]) may need to be reassessed to identify potential requirements for adjustment to ensure their sustainability. In the following sections, this article describes the features of the 2022-Recommendation, the revised NanoDefiner framework’s decision support flow scheme and e-tool, and standardisation developments that are important for its implementation. The FAIRification of expert knowledge within the e-tool’s knowledge base is addressed as well. The validity of the flow scheme and e-tool is confirmed via a broad set of revisited case studies, thus generating further guidance for the application of the 2022-Recommendation. The article closes with conclusions on the performed reassessment and provides perspectives for the updated and ready-to-use NanoDefiner framework.

## 2. Updated Nanomaterial Definition 2022/C 229/01 (2022-Recommendation) and Other Regulatory and Standardisation Considerations

The 2022-Recommendation is intended to serve different policy, legislative, and research purposes when addressing materials or issues concerning products of nanotechnologies [1]. Hence, in the near future, the 2022-Recommendation may be integrated in legislation, possibly with sector-specific adaptations. The 2022-Recommendation uses the $x_{50,0}$ as the main property for identifying a material as a nanomaterial, i.e., the median of the particle number-based distribution of particle external dimensions (“size”). In most cases, but not always, this refers to the smallest dimension. In the following, it will be called “$x_{50}$”. Furthermore, the volume-specific surface area (VSSA) [14,15], defined as the product of the specific surface area from Brunauer–Emmett–Teller (BET) measurements and skeletal density from gas pycnometry, is used as an exclusion criterion for nanomaterials. More precisely, a material with a specific surface area by volume smaller than $6\;m^{2}/{\mathrm{cm}}^{3}$ is not considered a nanomaterial. A general overview of available particle-sizing techniques was presented by [16].

For the regulatory implementation of the 2022-Recommendation, it should be possible to assess the required criteria using internationally accepted standardised test methods. This can be achieved by following two Organisation for Economic Co-operation and Development (OECD) Test Guidelines (TGs) that were adopted in June 2022: OECD TG 125 “Nanomaterial Particle Size and Size Distribution of Nanomaterials” [17] and OECD TG 124 “Volume Specific Surface Area of Manufactured Nanomaterials” [18]. These two OECD TGs close two major gaps regarding guidelines for testing the physicochemical properties of nanomaterials. A recent study [19] also clarifies which particles need to be counted, and how this should be carried out. Guidance on the new definition will clarify how to interpret the terms used in the 2022-Recommendation and assist applicants with information on how to assess a material against the required criteria. The 2022-Recommendation for the definition of a nanomaterial can thus be integrated and implemented in EU legislation, as already started for the EU Cosmetics Regulation (https://ec.europa.eu/growth/news/commission-seeks-views-eu-rules-cosmetic-products-2022-03-28_en (accessed 30 November 2022)). A comprehensive overview and in-depth discussion regarding these developments and further aspects are presented by [20], also providing further information on specific differences between the 2011-Recommendation and the 2022-Recommendation.

To prove that a material is not a nanomaterial, the 2011-Recommendation requires it to be shown that less than 50% of the particles in the particle number-based distribution have at least one external dimension between 1nm and 100nm. This becomes a major challenge when the particle size distribution is broad, with many particles considerably larger than 100nm. Therefore, empirical criteria and proxy methods were discussed in order to practically facilitate such an assessment [6,7,8,15,21]; however, there remains uncertainty as to whether such an assessment would be accepted in a regulatory context. With the introduction of a clear exclusion criterion based on VSSA, such empirical approaches are no longer needed in a regulatory context, which also simplifies the decision flow scheme for a strictly regulatory purpose. Such a decision flow scheme will be discussed in the next section. The empirical approaches for the categorisation “nanomaterial”/“not a nanomaterial” developed in the past will nevertheless still be useful, e.g., for the research and development of new materials, or for quality control, because these approaches are relatively fast and economic and are reasonably reliable. Furthermore, as legally binding definitions of nanomaterials based on the 2011-Recommendation are still in force, existing approaches to implement such definitions are still needed and valid. This is the case for the EU’s regulation on Registration, Evaluation, Authorisation and Restriction of Chemicals (REACH) [22]. The definition of “nanoform” in REACH was derived from the 2011-Recommendation. Moreover, the EU regulations on Biocidal Products and Medical Devices also contain definitions of a nanomaterial that are based on the 2011-Recommendation. Hence, it is appropriate to include in the NanoDefiner e-tool a new option (“regulatory purpose”) that may be used to fulfill new regulatory requirements. At the same time, it is appropriate to maintain the options that include empirical criteria, for the reasons explained above.

In recent years, considerable progress has been made by the International Organization for Standardization (ISO), with the publication of several standards dedicated to the measurement of particle size and shape distribution by electron microscopy. ISO 21363:2020 [23] describes in detail how to prepare the sample, how to capture electron micrographs, and how to analyse and report the data for transmission electron microscopy (TEM), based on seven case studies on representative (nano)materials. An analogue document is ISO 19479:2021 [24] for the case of the measurement of particle size and shape distribution by scanning electron microscopy (SEM). Even the 3D image reconstruction of nanoparticles by TEM has been developed and published as a technical specification, ISO/TS 22292:2021 [25]. These are just some examples of recent successful developments in the field of standardisation. Other ISO or European Committee for Standardization (CEN) projects on the determination of the (nano)particle size distribution with methods such as small-angle X-ray scattering (SAXS) and single-particle inductively coupled plasma mass spectrometry (spICP-MS), providing guidance on the measurement of the nanoparticle concentration or of the agglomeration/aggregation state, are in progress. Similarly, there is rich activity [26] under the pre-standardisation platform of the VAMAS project (http://www.vamas.org (access 30 November 2022)). This relies mostly on the organisation of international interlaboratory comparisons and provides valuable input for the development of new ISO standards on (nano)particle size, shape, and number concentration, with further methods (such as atomic force microscopy (AFM), also correlative with SEM), or new or more complex (nano)materials, e.g., 2D materials, including sample preparation. Last but not least, new reference nanoparticles with more complex size (non-spherical), shape distribution, and chemistry (core–shell structures) are in development at several institutes.

## 3. Updated NanoDefiner Flow Scheme Description

The updated decision support flow scheme, depicted in Figure 1, follows the 2022-Recommendation for the definition of a nanomaterial and uses its criteria in the various decision nodes. It avoids empirically derived proxy criteria, as it is unclear whether such criteria would be accepted for regulatory purposes. Such criteria might well remain meaningful for research, but they are not considered here for regulatory purposes.

Empirically derived proxy decision criteria for the assessment of whether a material fulfils the 2011-Recommendation, which go beyond regulatory requirements, were discussed earlier [9]. There, the VSSA was used as a proxy for the particle size distribution to identify materials that are not nanomaterials, although the 2011-Recommendation did not provide such an approach. It was pointed out that strictly applying the criteria of the 2011-Recommendation requires it to be explicitly shown that a material’s particle number-based distribution contains less than 50% of particles with external dimensions between 1nm and 100nm, which can be very challenging and may require expensive measurement procedures. The VSSA proxy approach was proposed earlier as an option to avoid such costly analysis. The 2022-Recommendation explicitly includes a criterion to identify materials that are not nanomaterials, which means that categorising a material as “not a nanomaterial” based on the VSSA is now possible for regulatory purposes. This is now integrated into the updated decision flow scheme.

Users start with the “basic categorisation” and determine whether the material in question belongs to any of the groups explicitly excluded from the 2022-Recommendation, i.e., whether it is non-particulate, non-solid particulate, or a nanostructured material. Further information on these terms is explained in [27] and will be revisited in upcoming guidance. If the material belongs to one of these groups, it can immediately be categorised as “not a nanomaterial”. If the material does not belong to these groups, one can continue with screening methods or alternatively go directly to the “Confirmatory step” (see Figure 1, “Confirmation” path from the “Strategy?” decision node). The “Screening” step is included here also for regulatory purposes, as it allows, under specific conditions, the categorisation of a material as a nanomaterial. Users who continue with screening can select the route of analysis at this stage. In order to be able to do this and select measurement methods that are appropriate for the material, the relevant physicochemical properties of the material should be known or obtained. Detailed information on how to select appropriate methods that are compatible with the material can be found in the NanoDefine Methods Manual [6,7,8]. If the material is in liquid dispersion, users should select the “Dispersion route” in the screening step. If the “Dispersion route” is selected for a material present in dry powder form, users should carefully apply an appropriate dispersion method. The particle size distribution resulting from a screening method in the dispersion route can only be used to identify a material as nanomaterial if the median of the distribution is smaller than or equal to 100nm. That is why screening methods are included in the dispersion route of the flow scheme. However, it cannot be used to identify a material as “not a nanomaterial” if the median is larger than 100 nm. In the latter case, a confirmatory step is needed. If the material is in powder form, users might want to select the powder route first and proceed with the measurement of the volume-specific surface area (VSSA). If the determined VSSA is smaller than $6\;m^{2}/{\mathrm{cm}}^{3}$, the material fulfils the VSSA criterion of the definition and can be categorised as “not a nanomaterial”. If the VSSA is greater than or equal to $6\;m^{2}/{\mathrm{cm}}^{3}$, the confirmatory step is required. The direct categorisation of materials that are not nanomaterials via VSSA was introduced with the new definition; this was not possible before [2].

The confirmatory step is needed if users decide to skip the screening or if no categorisation is possible in the screening step. The outcome of the confirmatory step is then used in the decision to categorise something as “nanomaterial” or “not a nanomaterial”.

## 4. Updated NanoDefiner e-Tool and FAIRification

The NanoDefiner e-tool (https://github.com/NanoDefiner/NanoDefiner (accessed 30 November 2022); https://zenodo.org/record/7607457 (accessed 5 February 2023)) [6,10] is an expert system that supports users with different levels of expertise in the process of nanomaterial categorisation, according to the 2011- and 2021-Recommendations. It implements a flow-scheme-based [6,9] and guided workflow that enables dossier creation for regulatory (e.g., REACH) or scientific assessment purposes. This comprises (i) a detailed description of particulate components in a sample via a material categorisation scheme [6,28], (ii) the recommendation and selection of adequate measurement techniques [7,10], and (iii) the processing and compilation of the analysis results (automatically for ParticleSizer (https://www.imagej.net/ParticleSizer/ (accessed 30 November 2022)) [29] data, Single Particle Calculation tool (https://www.wur.nl/en/show/Single-Particle-Calculation-tool.htm (accessed 30 November 2022)) [30] data, and VSSA), and (iv) report generation as Appendix A for documentation and registration. Recommendations and decisions are explained by the e-tool, yet inputs and imports are not questioned to enable a broad applicability. Hence, users remain responsible for the correctness of all input provided to the e-tool.

A public version was released in 2017 as open-source software along with a try-out service (https://labs.inf.fh-dortmund.de/NanoDefiner/ (access 30 November 2022)) for interested users, rated “A” (i.e., very good) in a Qualys Secure Sockets Layer (SSL) analysis (https://www.ssllabs.com/ssltest/ (accessed 30 November 2022)). As of December 2022, there have been more than 170 service registrations from industry and academia, more than 160 GitHub release downloads (https://githubstats0.firebaseapp.com/ (accessed 30 November 2022)), not including repository clones, and more than 40 Docker image downloads (https://hub.docker.com/v2/repositories/nanodefiner/nanodefiner (accessed 30 November 2022)) were recognised, suggesting considerable stakeholder interest. Three updates of the e-tool have been released since its initial release. These have provided (i) portable document format (PDF)/A-1a reports for long-term archiving, (ii) terminology and manual updates according to the concepts and terms of the 2011-Recommendation [27], and (iii) bug fixes. Continuous updates of crucial dependencies to protect against discovered security vulnerabilities have been conducted as well. In addition to further security updates, the e-tool’s current fourth update includes two major novelties: the adoption of the updated decision support flow scheme according to the 2022-Recommendation described in Section 3, and the FAIRification of its foundation, i.e., the knowledge base.

The term FAIRification refers to the Findable, Accessible, Interoperable, and Reusable (FAIR) principles (https://www.go-fair.org/fair-principles/ (accessed 30 November 2022)) [31]. These principles address basic but crucial requirements for scientific data to be, in the widest sense of the word, sustainable. The FAIR principles have been composed under the light of the infrastructural flaws in an ever-growing system of research and development that hinder or complicate reuse. A consortium of stakeholders from academia, industry, funding, and publishing was involved in the development. The resulting principles are intended as guidelines with measurable impact, specifically targeting the facilitation of automatic data discovery and usage. However, the support of reuse by individuals is also addressed. The importance of following these principles for scientific data in the field of nanosafety research has recently been discussed [32]. Data gathered over years by large consortia with considerable funding are often not reusable without vast administrative effort.

The e-tool’s knowledge base holds formalised expert knowledge on the material properties and performance characteristics of measurement techniques. Recommendations from the e-tool on measurement techniques are based on matching. Data were aggregated throughout the NanoDefine project and published in the NanoDefine Methods Manual [6,7,8], but not in a machine-processable manner. The FAIRified data present various opportunities for all stakeholders, namely users from industry and academia, as well as authorities. The decisions of the e-tool are already explained within the user interface and comprehensible on the lowest possible level, allowing the traceability of potential irregularities. This aids research and is especially relevant for industry and academia, aiming to compose and test custom material property and measurement technique performance profiles. Such profiles may be provided publicly for knowledge base extension purposes, e.g., for novel or manufacturer-specific reference materials and measurement devices. Authorities gain an additional way to validate supplementary dossier reports generated by the e-tool, as the used knowledge base versions and profiles are now listed and thus verifiable against public records of it. Aspects of how the knowledge base’s FAIRification was implemented are reported concisely in Table 1.

## 5. Selected Case Studies Revisited under the 2022-Recommendation

Material characterisation data and images of five materials are revisited and reported in the following. Data acquisition and analysis were conducted within the NanoDefine project according to published standard operating procedures [8]. All data used in the following case studies are available through NanoDefines publications, e.g., in an overview-table [33]. The samples comprise the materials IRMM-389 (basic methacrylate copolymer), IRMM-387 (${\mathrm{BaSO}}_{4}$, ultrafine grade), IRMM-380 (pigment yellow 83 nano), IRMM-384 (${\mathrm{CaCO}}_{3}$, fine grade), and IRMM-382 (multi-walled carbon nanotubes). SEM/TEM micrographs of these materials are shown in Figure 2a, Figure 3a, Figure 4a, Figure 5a, Figure 6a and Figure 7a, respectively. The purpose of this material selection was to challenge all branches of the updated flow scheme, shown in Figure 1, as well as the updated e-tool. Appendix A comprise dossier reports generated by the e-tool.

The characterisation of an unknown material usually follows the tiered approach, as developed within the NanoDefine project. The basic categorisation starts with collecting a priori information on the respective material, e.g., from the manufacturer specifications. In many cases, descriptive SEM or VSSA data are already available for materials to be assessed against the criteria of the 2022-Recommendation. One basic objective of the flow scheme is to come to a decision with cost-efficient screening “tier 1” techniques and only use expensive confirmatory “tier 2” techniques in borderline cases, i.e., when the results of tier 1 schemes are inconclusive. Details on the revisited case studies are reported in the following subsections.

The results clearly show that the updated, simplified flow scheme leads to a streamlined categorisation strategy. From a user point of view, it is important to be aware of the fact that the flow scheme only covers the requirements of the EC recommendation for a nanomaterial definition. A user, e.g., from industry, seeking material registration for regulatory purposes should consider further aspects:In many cases, tier 1 results are already available and therefore the analyst has prior knowledge on, for example, the particle size range.In the case of a categorisation of “nanomaterial”, the European Chemicals Agency (ECHA) requires considerably more data than just the median of the particle number-based size distribution, $x_{50}$, for a dossier intended for registration according to REACH.Some analysis techniques deliver more parameters than just a $x_{50}$, which could change their impact-to-cost ratio.

Taking the above points into consideration, for materials available as powders, in most cases it is advisable to start with a determination of the VSSA derived from the results obtained from the BET method and from skeletal density provided by gas pycnometry. VSSA is the only criterion with the potential to directly categorise a material as “not a nanomaterial” and, therefore, further analysis of the categorisation of the material is avoided. In most cases, this information is already available and, in such a case, requires no further analysis. Categorisation as “nanomaterial” based on the VSSA is not possible according to the 2022-Recommendation, even in the case of a $\mathrm{VSSA}\geq 6\;m^{2}/{\mathrm{cm}}^{3}$. Other flow schemes and proposals involving, for example, a shape criterion [15,34] are therefore not applicable in the context of the 2022-Recommendation.

Categorisation as “nanomaterial” using only tier 1 methods is possible, as shown in some selected case studies in the following. However, REACH registration also requires the reporting of the aspect ratio, shape, crystallinity, rigidity, and many more particle parameters. Many of these requirements can be fulfilled through information obtained from the appropriate evaluation of electron microscopy images. It is therefore, in many cases, the best strategy to directly use SEM or TEM in the case of a $\mathrm{VSSA}\geq 6\;m^{2}/{\mathrm{cm}}^{3}$. The data used for the present cases were also entered in the updated e-tool version to test the consistency of the decision-making framework. The corresponding dossiers generated by the e-tool are available as supportive information. The reports consist of a larger collection of sizing methods than that presented in the case studies.

### 5.1. Case Study on IRMM-389 (Basic Methacrylate Copolymer)

Figure 2 describes the easiest case of material categorisation. IRMM-389 (basic methacrylate copolymer) consists of large particles in the size range of several µm, resulting in a low $\mathrm{VSSA}=1\;m^{2}/{\mathrm{cm}}^{3}$. The SEM micrograph shows a representative selection of the sample. The particles have an irregular shape with a small aspect ratio and a low degree of agglomeration. The updated recommendation allows for direct categorisation as “not a nanomaterial”, based on the $\mathrm{VSSA}<6\;m^{2}/{\mathrm{cm}}^{3}$. This was already proposed by the original NanoDefine decision flow scheme, but not covered by the 2011-Recommendation.
Figure 2IRMM-389 (basic methacrylate copolymer): (**a**) representative SEM micrograph, and (**b**) respective flow scheme path. The tier 1 method BET yielding a VSSA=1m2/cm3 directly leads to a “not a nanomaterial” categorisation.
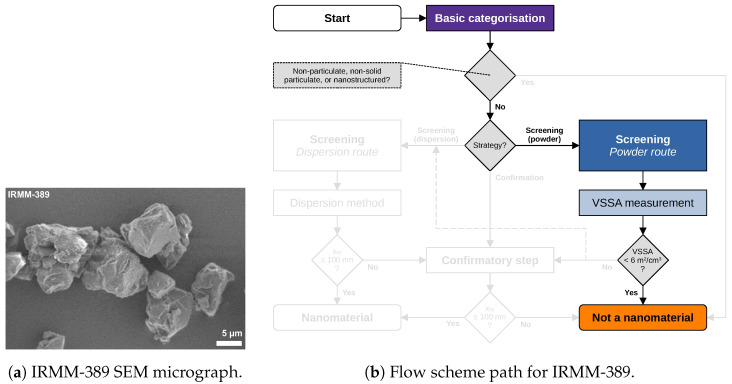



### 5.2. Case Study on IRMM-387 (BaSO_4_, Ultrafine Grade)

Figure 3 describes another case study of IRMM-387 (${\mathrm{BaSO}}_{4}$, ultrafine grade), where the tier 1 method dynamic light scattering (DLS) directly leads to the categorisation of “nanomaterial” with an $x_{50}=76\;\mathrm{nm}$. Tier 1 methods of the dispersion route only require a confirmatory step in the case of an $x_{50}>100\;\mathrm{nm}$, in order to avoid false-negative categorisation. The categorisation as “nanomaterial” by DLS was checked by SEM, with the results showing that this tier 1 categorisation is correct. IRMM-387 shows strong aggregation. Sample preparation for DLS therefore had to ensure appropriate dispersion by following the dispersion protocols provided by the NanoDefine project [35].
Figure 3IRMM-387 (BaSO4, ultrafine grade): (**a**) representative SEM micrograph, and (**b**) respective flow scheme path. The tier 1 method DLS yielding an x50=76nm directly leads to a “nanomaterial” categorisation that was later checked as correct by SEM.
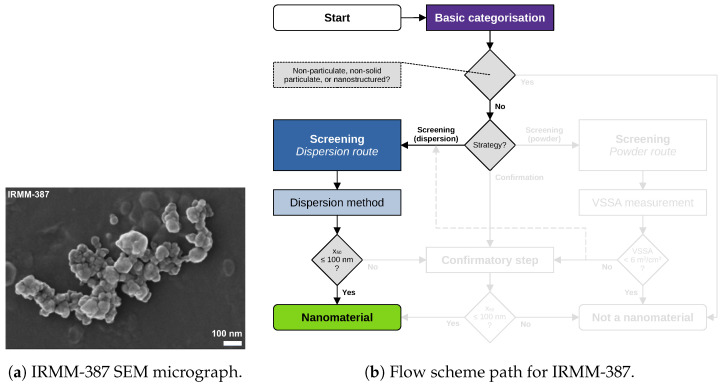



Figure 4 shows another possible path in the flow scheme for the same material IRMM-387. BET was selected as a first step, resulting in a rather high $\mathrm{VSSA}=149\;m^{2}/{\mathrm{cm}}^{3}$. Such a VSSA is a strong indication for a nanomaterial. However, categorisation as a “nanomaterial” by BET is not allowed by the 2022-Recommendation. Therefore, the user chooses analysis by a second tier 1 method (e.g., the route which is already described by Figure 3). For IRMM-387 analyses, additional tier 1 methods, e.g., centrifugation-based ones, were also performed. Each method consistently results in a “nanomaterial” categorisation.
Figure 4IRMM-387 (BaSO4, ultrafine grade): (**a**) representative SEM micrograph, and (**b**) respective flow scheme path. First, the tier 1 method BET yielding a VSSA=149m2/cm3 leads to a switch from the powder route to the dispersion route. Then, the tier 1 method DLS yielding an x50=76nm leads to a “nanomaterial” categorisation.
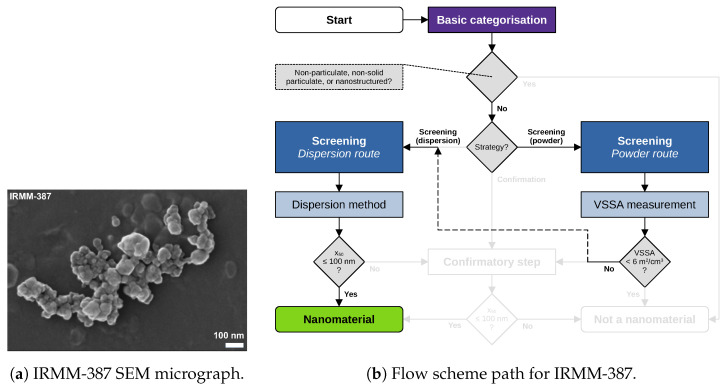



### 5.3. Case Study on IRMM-380 (Pigment Yellow 83 Nano)

Figure 5 describes a very common flow scheme. The SEM micrograph shows that IRMM-380 (pigment yellow 83 nano) most likely consists of nanoparticles. The high $\mathrm{VSSA}=93\;m^{2}/{\mathrm{cm}}^{3}$ cannot be used for a direct decision (note that only a $\mathrm{VSSA}<6\;m^{2}/{\mathrm{cm}}^{3}$ is allowed to be used to categorise a material as “not a nanomaterial” according to the 2022-Recommendation). Therefore, it was in this case combined with the confirmatory method TEM, which provided an $x_{50}=39\;\mathrm{nm}$ (for the smallest dimension) and, hence, a clear categorisation as “nanomaterial”.
Figure 5IRMM-380 (pigment yellow 83 nano): (**a**) representative SEM micrograph, and (**b**) respective flow scheme path. First, the tier 1 method BET yielding a VSSA=93m2/cm3 leads to the decision to perform a confirmatory step. Then, the tier 2 method TEM yielding an x50=39nm leads to a “nanomaterial” categorisation.
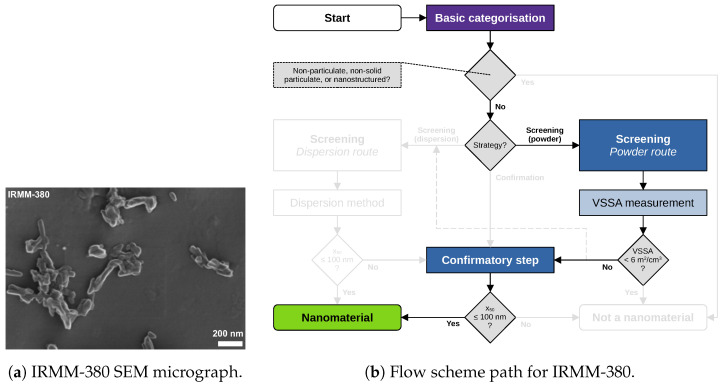



### 5.4. Case Study on IRMM-384 (CaCO_3_, Fine Grade)

Figure 6 describes the case of IRMM-384 (${\mathrm{CaCO}}_{3}$, fine grade), which has a $\mathrm{VSSA}=15\;m^{2}/{\mathrm{cm}}^{3}$. While such a medium VSSA does not lead to a direct categorisation as “not a nanomaterial”, the confirmatory TEM results in an $x_{50}=153\;\mathrm{nm}$ (for the smallest dimension) and, therefore, unambiguous categorisation as “not a nanomaterial”. The TEM micrograph shows an increased aspect ratio of the particles. While the NanoDefine project proposed shape-dependent adaptations to the VSSA calculation [15], which would result in a direct decision as “not a nanomaterial” for IRMM-384, this approach is not included in the updated 2022-Recommendation. The case study therefore demonstrates a main difference compared with the previous decision flow scheme [6,9], also reflected in the preceding e-tool release.
Figure 6IRMM-384 (CaCO3, fine grade): (**a**) representative TEM micrograph, and (**b**) the respective flow scheme path. The tier 1 method BET yielding a VSSA=15m2/cm3 does not necessarily lead to a “nanomaterial” categorisation in a subsequent confirmatory step. The following tier 2 method TEM yielding an x50=153nm leads to a “not a nanomaterial” categorisation.
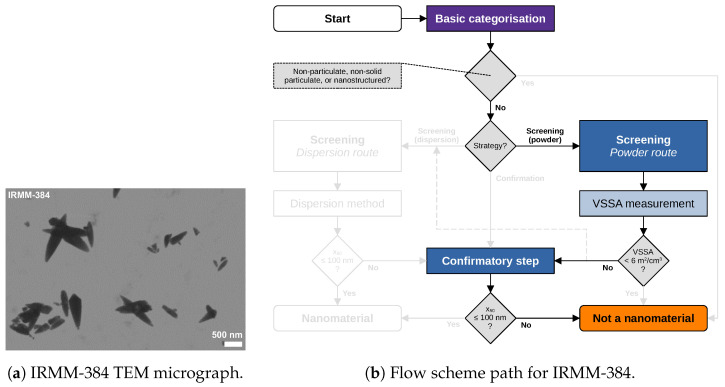



### 5.5. Case Study on IRMM-382 (Multi-Walled Carbon Nanotubes)

Carbon-based nanomaterials, e.g., carbon nanotubes, are separately addressed in the 2011-Recommendation and, together with other particles of elongated shape, are implicitly included in the 2022-Recommendation. They are covered by the nanomaterial definitions, even if they have a minimum Feret diameter below the 1nm threshold combined with a very high aspect ratio. The flow scheme therefore also applies to this group of materials, as shown in Figure 7. The present material IRMM-382 (multi-walled carbon nanotubes) was analysed via the BET method, resulting in a $\mathrm{VSSA}=480\;m^{2}/{\mathrm{cm}}^{3}$. This is a clear indication that the particles are very small. Even so, BET does not result in a direct categorisation of this material as a “nanomaterial”, as this criterion is no longer present in the 2022-Recommendation. Thus, TEM was used as the confirmatory step. The smallest dimension determined by TEM is an $x_{50}=12\;\mathrm{nm}$, resulting in categorisation as a “nanomaterial”. This case study is also an example of the efficiency of the workflow being the same when skipping the tier 1 screening. Multi-walled carbon nanotube materials are not designed to have a $\mathrm{VSSA}<6\;m^{2}/{\mathrm{cm}}^{3}$, and are not easily dispersed; hence, the user may prefer to go directly to the tier 2 confirmatory step. However, if BET data are available, as is frequently the case, no time is lost in the tier 1 screening step.
Figure 7IRMM-382 (multi-walled carbon nanotubes): (**a**) representative SEM micrograph, and (**b**) the respective flow scheme path. The tier 1 BET method yielding a considerably large VSSA=480m2/cm3 makes it viable to skip a switch to the dispersion route using other screening tier 1 methods and continue with a subsequent confirmatory step. The following tier 2 method TEM with an x50=12nm leads to a “nanomaterial” categorisation.
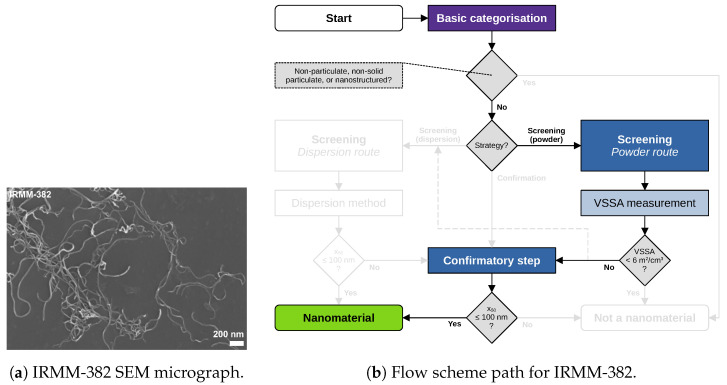



## 6. Conclusions and Perspectives

The EC’s recommendation for the definition of “nanomaterial” was updated in 2022, with the purpose to improve its clarity and to achieve a better implementability in a regulatory context. The resulting 2022-Recommendation has clearer wording and uses more precise terms. It allows for an easier assessment of materials whether they fulfil the definition or not, as compared to the previous 2011-Recommendation. This can be visualised in a simpler decision tree, which avoids proxy criteria and includes a clear criterion to identify materials that are not nanomaterials, based on the VSSA and applicable to materials in dry powder form.

The updated flow scheme and free, open-source e-tool allow users to assess materials specifically against the criteria of the 2022-Recommendation, and to decide whether they are to be categorised as nanomaterials or not. In parallel, it continues supporting material categorisation according to the NanoDefine approach, depicted in the previous flow scheme [6,9]. Ongoing maintenance ensures bug fixes and security updates. The FAIRified knowledge base creates additional possibilities to validate e-tool decisions and enables extension with custom materials and measurement technique profiles by researchers and manufacturers. Reuse is granted in a broad context due to publication under the industry-friendly MIT license.

The revisited case studies according to the updated flow scheme demonstrate a simplified analytical approach, as VSSA by BET can lead to a fast and cost-efficient categorisation as “not a nanomaterial” (see Section 5.1, and compare Section 5.4 to [9]). Despite this simplification, no deviations in categorisations compared to the 2011-Recommendation and corresponding e-tool implementation were observed. It was demonstrated that, in many cases, a decision can be made by employing screening tier 1 methods only. Yet, the REACH registration of a nanomaterial requires additional information, which can only be provided by confirmatory tier 2 methods. The demonstrated case-study-driven approach may serve as an example of how to revisit and validate other decision support tools for nanomaterial definition-related questions [13], initially developed for the 2011-Recommendation. Remaining challenges include plate-like materials with a high degree of agglomeration/aggregation, as these require individual approaches for the confirmatory tier 2 methods. The e-tool is designed to provide recommendations for such cases with difficult boundary conditions, and further guidance on the identification of nanomaterials through measurements [21] is available.

Activities for harmonising [20] the definition of nanomaterials across different legislative sectors in the EU (chemicals (REACH), novel foods, cosmetic products, biocides, and medical devices) are ongoing with the 2022-Recommendation as the starting point. This is in line with the EC’s “one substance, one assessment” vision for chemicals. Once the nanomaterial definitions are harmonised, the flow scheme and e-tool can easily be adapted to possible sector-specific modifications of the definition. Such adaptations can be implemented in the flow scheme and e-tool through options selectable by the user. As a result, the task of categorising nanomaterials across sectors will be possible through a single tool, greatly simplifying specific regulatory obligations.

### Disclaimer

The content expressed in this paper is solely the opinion of the authors and does not necessarily reflect the opinions of their institutions. The contribution of the author A. Mech to the paper was made before she joined EFSA. The author A. Mech is currently employed with the EFSA in the Food Ingredients and Packaging unit, which provides scientific and administrative support in the area of food additives and packaging safety. However, the present article is published under the sole responsibility of the author and may not be considered as an EFSA scientific output. The positions and opinions presented in this article are those of the author alone and are not intended to represent the views or any official position or scientific work of EFSA. To know the views or scientific outputs of EFSA, please consult its website under http://www.efsa.europa.eu.

## Figures and Tables

**Figure 1 nanomaterials-13-00990-f001:**
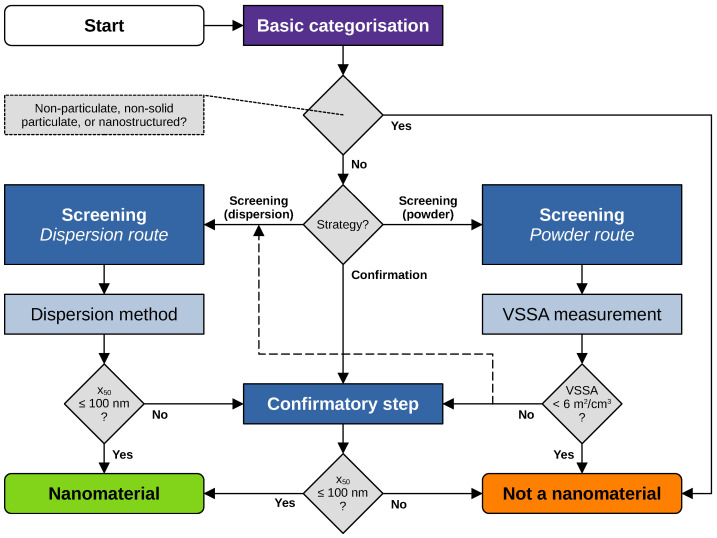
Updated analysis and decision flow scheme following the 2022-Recommendation for the definition of a nanomaterial. $x_{50}$ is the median of the particle number-based size distribution, used here as short form for $x_{50,0}$.

**Table 1 nanomaterials-13-00990-t001:** FAIR principles and their implementation for the NanoDefiner e-tool knowledge base.

Principle	Description, Taken from [31]	Implementation
*Findable*		
F1	(Meta)data are assigned a globally unique and persistent identifier.	The NanoDefiner e-tool GitHub repository holding the knowledge base was registered with Zenodo and has received a respective Digital Object Identifier (DOI): 10.5281/zenodo.7607457
F2	Data are described with rich metadata (defined by R1 below).	Specific profile data are described with context-rich attributes derived from the NanoDefine Methods Manual.
F3	Metadata clearly and explicitly include the identifier of the data they describe.	The Zenodo DOI is used to link the metadata and the knowledge base.
F4	(Meta)data are registered or indexed in a searchable resource.	The knowledge base is findable via search engines as well as directly on Zenodo and GitHub.
*Accessible*		
A1	(Meta)data are retrievable by their identifier using a standardised communications protocol.	The knowledge base is accessible via Hypertext Transfer Protocol Secure (HTTPS) on GitHub and Zenodo.
A1.1	The protocol is open, free, and universally implementable.	The HTTPS protocol is open, free, and universally implementable.
A1.2	The protocol allows for an authentication and authorisation procedure, where necessary.	The HTTPS protocol allows for an authentication and authorisation procedure where necessary.
A2	Metadata are accessible, even when the data are no longer available.	Registration with Zenodo ensures availability of metadata, even if the GitHub repository holding the knowledge base is no longer available, and vice versa.
*Interoperable*		
I1	(Meta)data use a formal, accessible, shared, and broadly applicable language for knowledge representation.	The knowledge base uses plain-text comma-separated values (CSV) and is further provided as Excel Microsoft Office Open XML (XSLX).
I2	(Meta)data use vocabularies that follow FAIR principles	Not applicable.
I3	(Meta)data include qualified references to other (meta)data.	The knowledge base includes references to the NanoDefine Methods Manual.
*Reusable*		
R1	(Meta)data are richly described with a plurality of accurate and relevant attributes.	The knowledge base contains a large number of context-rich attributes derived from the NanoDefine Methods Manual for profile data description.
R1.1	(Meta)data are released with a clear and accessible data usage license.	The NanoDefiner e-tool and its knowledge base are released under the Massachusetts Institute of Technology (MIT) license.
R1.2	(Meta)data are associated with detailed provenance.	Detailed provenance is provided in the NanoDefine Methods Manual [6,7,8].
R1.3	(Meta)data meet domain-relevant community standards.	The knowledge base was compiled with the help of domain experts, revised and checked for consistency over several iterations [10], and meets community standards.

## Data Availability

Further data on material analyses are available upon request from the corresponding author. The NanoDefiner e-tool source code and knowledge base are available at the official GitHub repository (https://github.com/NanoDefiner/NanoDefiner (accessed 30 November 2022)) and have further been registered with Zenodo (https://doi.org/10.5281/zenodo.7607457 (accessed 5 February 2023)) (Concept DOI referring to the latest release).

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
