# Peer review of "NanoDefiner Framework and e-Tool Revisited According to the European Commission’s Nanomaterial Definition 2022/C 229/01"

_nanomaterials, 2023, doi:10.3390/nano13060990_

Round 1

Reviewer 1 Report

The authors aim to present an article on NanoDefiner Framework and e-Tool Revisited According to the European Commission’s Nanomaterial Definition 2022/C 229/01. This paper highlights the important aspects mainly features of the 2022-Recommendation, the revised NanoDefiner framework’s decision support flow scheme and e-tool, and standardization developments important for its implementation. The manuscript is well written, and the figures are also presented in a nice manner having a lot of new information. Before accepting the manuscript, authors are advised to address a few points:

1.        Please include a short (~35 word) teaser – this should be a short summary statement intended to whet the appetite of the reader.

2.      The authors may briefly describe how they went about data collection prior to writing the article. This would add more value to the work.

3.      Reviewers encourage authors to include some case studies or some more literature for the specificity of the article.

4.      Formatting and grammatical errors should be verified again throughout the manuscript.

Reviewer 2 Report

The authors introduced important regulation in this article and this information seems to be quite helpful for researchers and industry. So, I would like to suggest minor revision before acceptance of article.

1. The authors emphasize the values about volume specific surface area (VSSA) and X50. So, please explain those terms more clearly for better understanding and introduce measurement methods of each property.

2. As the authors knew it, there are lots of types of nanomaterials not only polymeric nanoparticle or inorganic nanoparticle but also carbon nanomaterials such as carbon nanotubes, graphene, or 2D layered structures and so on. And some of them, they have been applied for the various industrial fields. So, please add the examples including such carbon nanomaterials and 2D layered structure as case study.
